# Spatiotemporal Mobility Based Trajectory Privacy-Preserving Algorithm in Location-Based Services

**DOI:** 10.3390/s21062021

**Published:** 2021-03-12

**Authors:** Zhiping Xu, Jing Zhang, Pei-wei Tsai, Liwei Lin, Chao Zhuo

**Affiliations:** 1School of Computer Science and Mathematics, Fujian University of Technology, Fuzhou 350118, China; czndxu@163.com (Z.X.); llw02_02@163.com (L.L.); aryj2021@163.com (C.Z.); 2Fujian Provincial Key Laboratory of Big Data Mining and Applications, Fujian University of Technology, Fuzhou 350118, China; 3Department of Computer Science and Software Engineering, Swinburne University of Technology, Hawthorn 3122, Australia; pwtsai@foxmail.com

**Keywords:** location-based services, trajectory privacy, trajectory data publishing, *k*-anonymity, spatiotemporal mobility

## Abstract

Recent years have seen the wide application of Location-Based Services (LBSs) in our daily life. Although users can enjoy many conveniences from the LBSs, they may lose their trajectory privacy when their location data are collected. Therefore, it is urgent to protect the user’s trajectory privacy while providing high quality services. Trajectory *k*-anonymity is one of the most important technologies to protect the user’s trajectory privacy. However, the user’s attributes are rarely considered when constructing the *k*-anonymity set. It results in that the user’s trajectories are especially vulnerable. To solve the problem, in this paper, a Spatiotemporal Mobility (SM) measurement is defined for calculating the relationship between the user’s attributes and the anonymity set. Furthermore, a trajectory graph is designed to model the relationship between trajectories. Based on the user’s attributes and the trajectory graph, the SM based trajectory privacy-preserving algorithm (MTPPA) is proposed. The optimal *k*-anonymity set is obtained by the simulated annealing algorithm. The experimental results show that the privacy disclosure probability of the anonymity set obtained by MTPPA is about 40% lower than those obtained by the existing algorithms while the same quality of services can be provided.

## 1. Introduction

As enabled by the maturity of 5G technologies, location-based services (LBSs) have become popular in our daily life [1]. However, these service providers have stored the user’s trajectory data [2]. The trajectory data contains a large amount of the user’s sensitive information, such as shopping habits, home address, workplace, or frequently visited places [3]. If these service providers suffer from security breaches or the data flow is used by attackers maliciously, the trajectory data may be directly leaked without any protection. It would result in exposing the sensitive information regarding the user. Therefore, finding a way to protect the user’s trajectory data for better privacy is necessary.

In response to the need mentioned above, researchers have worked extensively on related trajectory privacy protection technologies [4]. k-anonymity is one of the important techniques recently used to protect a user’s trajectory. The k-anonymity set is formed by similar trajectories and sent to the service providers [5], where k denotes the anonymity degree. Nevertheless, constructing a good *k*-anonymity set effectively is a big challenge because the attacker may consider side information and use data mining techniques to distinguish the dummy trajectories.

For constructing the k-anonymity set, most of the existing approaches consider the direction similarity between trajectories [6,7,8,9,10,11,12,13,14,15,16]. However, these methods ignore that different users have different attributes and movement patterns. The trajectories generated by different attributes of users are very different.

In this paper, Spatiotemporal Mobility (SM) is used to denote the user’s attributes with respect to the number of stopovers and the average moving speed of the user. The stopovers include the supermarket, the park, the community, or any locations the user may visit. The attacker can still distinguish the trajectory in the anonymity set through the SM. Figure 1 shows two users’ moving trajectories in one day. The trajectories colored in red and green belong to Alice and Bob, respectively. The stopovers of Alice are distributed over multiple locations in the region. Her average moving speed is comparatively high. It is easy to speculate that her daily movement pattern is irregular and unfixed. On the contrary, the stopovers of Bob only distribute in two locations. He has fewer stopovers than Alice. His average moving speed is lower. It is speculated that his daily movement pattern is likely to be more regular and fixed. It is concluded that the mobility of Alice is higher than Bob. If the trajectory k-anonymity set submitted by Bob contains a trajectory generated by Alice, once the attacker knows Bob is an employee of a company through data mining techniques, this trajectory with high mobility in the anonymity set will be easily filtered out.

Motivated by the above, this paper aims to explore the way to construct a k-anonymity set. There are two issues to be considered. The first is how to measure the similarity between trajectories. The second one is how to make the trajectories in the *k*-anonymity set more similar. To address these two issues, a novel trajectory privacy-preserving algorithm is proposed. The main contributions of this paper are listed as follows:

The SM is defined based on the number of stopovers and the average moving speed of the user’s trajectory. Furthermore, the SM is used to measure the similarity between trajectories to form the trajectory k-anonymity set.The trajectory graph is constructed to model the relationship between trajectories. The analysis of the relationship between trajectories is transformed into the study of graph features.The Spatiotemporal Mobility-based Trajectory Privacy-Preserving Algorithm (MTPPA) is proposed. The k-anonymity set is constructed by the historical trajectories with the simulated annealing algorithm. This anonymity set improves the similarity between the anonymity set trajectories effectively.The performances are analyzed by the real datasets [6]. The results show that the k-anonymity set constructed by MTPPA has a lower trajectory privacy disclosure prob-ability than existing algorithms while ensuring the quality of services.

The remainder of this article is organized as follows. The related works are discussed in Section 2. The problem formulation is explained in Section 3. The proposed MTPPA is revealed in Section 4. The experimental results and analysis are delivered in Section 5. Finally, the conclusion is given in Section 6.

## 2. Related Works

As one of the most important trajectory privacy protection technologies, the k-anonymity method was proposed by Gruteser et al. [7] in 2003. The anonymity set is constructed with k similar trajectories. The probability of an attacker distinguishing a particular user is less than 1/k. There are three kinds of approaches based on the k-anonymity method: the dummy trajectory method, the suppression method and the generalization method.

The dummy trajectory method generates k−1 similar dummy trajectories to form the k-anonymity set. When generating k−1 similar trajectories, Liu et al. [8] select the final anonymity set from three aspects including the time reachability, the direction similarity, and the in-degree/out-degree. Wang et al. [9] rotate the user’s real trajectory at the selected rotation point to generate k−1 dummy trajectories. Shaham et al. [10] select k−1 dummy locations with the same posterior probability of the real location. The transfer probability of each location to the next k-anonymity set is equal. They generate multiple dummy location sets and divide them into several subsets. Then, they select the anonymity set which has the largest entropy. However, the above methods do not meet the requirements of real geographical constraints in most cases.

The suppression method constructs the k-anonymity set by removing the highly sensitive locations from the trajectory collection. Zhao et al. [11] suppress the whole problematic trajectory data locally according to the trajectory frequency and the relationship between privacy relevance and data utility. To construct the k-anonymity set, Gramaglia et al. [12] suppress the sampling points so that the data spatiotemporal granularity is minimized. Li et al. [13] use the hidden Markov model to formulate the user’s mobile status and the visited locations. A probability vector of the user’s mobile direction is used as the decision variable to determine whether revealing the user’s trajectory details. However, these methods lead to excessive trajectory information loss.

The generalization method generalizes a trajectory into a k-anonymity set. Each record of the location at a timestamp is a generalized region. Based on the traditional generalization method, Xu et al. [14] consider four characteristics of direction, speed, time and space as the basis for measuring the similarity of trajectories. Xin et al. [15] use the Gibbs sampling clustering method to detect the representative regions. Then, the detected representative regions are further generalized according to the rationality of equivalence classes. Zhang et al. [16] propose a trilateral Stackelberg game model based on community structure. They design an optimization method to construct the k-anonymity set by the reverse induction method. However, when the road network is too sparse, the anonymous region of the above methods is also large.

To generate dummy trajectories that match the real geographical constraints, in this paper, the historical trajectories are used to construct the k-anonymity set. Furthermore, the SM is used to measure the similarity between trajectories. Trajectories with similar mobility level make it more difficult for the attacker to distinguish the trajectories.

## 3. Problem Formulation

The basic properties and relations used in this paper are briefly reviewed in this section. The important symbols with their definitions are shown in Table 1.

**Definition** **1.***Trajectory* [17]. *The user’s trajectory*
T
*is considered as a polyline in the three-dimensional space. It is composed of a sequence of sampling points accessed over time. Hence,*
T
*is defined as follows:*$$T=\left\{\left(x_{1},\;y_{1},t_{1}\right),\left(x_{2},\;y_{2},t_{2}\right),\ldots ,\left(x_{i},\;y_{i},t_{i}\right),\ldots ,\left(x_{n},\;y_{n},t_{n}\right)\right\},$$
*where, a sampling point*
$\left(x_{i},\;y_{i},t_{i}\right)$
*represents the user’s coordinate*
$\left(x_{i},\;y_{i}\right)$
*at sampling time*
$t_{i}$.
*Given a starting timestamp*
$t_{s}$
*and an ending timestamp*
$t_{e}$
*, two trajectories*
$T_{i}$
*and*
$T_{j}$
*are extracted to form an equivalence class when all of their sampling points are in the same time interval*
$[t_{s}$
*,*
$t_{e}]$
*[14]. If*
$T_{i}$
*and*
$T_{j}$
*of an equivalence class have the same number of sampling points in the same sampling time length, they are synchronized trajectories [18]. A trajectories set is called a synchronized trajectory set if any two trajectories from the set are synchronized.*


**Definition** **2.***Stopover* [18]. *The stopover*
S
*of a user refers to a specific site or place (e.g., a bus station, a market, or even the user’s homesite) where the location is functional, useful, or meaningful to the user.*

**Definition** **3.**
*Spatiotemporal mobility. The spatiotemporal mobility*
M
*of a user is measured by the sum of the number of stopovers*
N
*and the average moving speed*
$\bar{v}$
*of the user’s trajectory. In the time interval [*
$t_{1}$
*,*
$t_{n}$
*], the average moving speed*
$\bar{v}$
*is the ratio of the total length of the trajectory to the total moving time, which is presented in Equation (1):*
(1)$$\bar{v}=\frac{\sum_{i=1}^{n-1}\sqrt{{\left(x_{i+1}-x_{i}\right)}^{2}+{\left(y_{i+1}-y_{i}\right)}^{2}}}{t_{n}-t_{1}}.$$
*After applying the normalization process, the SM of a trajectory is defined as follows:*(2)$$M=\alpha \frac{N}{n}+\beta \frac{\bar{v}}{v_{max}},$$*where,*$v_{max}$*is the maximum speed limit of the anonymous region,*n*is the number of sampling points,*α*and*β*represent the proportion of the number of stopovers and the average moving speed of the SM, respectively.*α,β∈[0,1]*and*α+β=1. 

**Definition** **4.**
*Trajectory Similarity. The SM difference between two synchronized trajectories is used to measure the trajectory similarity.*
*Let the SM of synchronized trajectories*$T_{i}$*and*$T_{j}$*generated by two users called*$M_{i}$*and*$M_{j}$*. The mobility difference between*$T_{i}$*and*$T_{j}$*is defined as the absolute value of the difference between*$M_{i}$*and*$M_{j}$*and given as follows:*(3)$$\Delta M\left(T_{i},T_{j}\right)=\left|M_{i}-M_{j}\right|,$$*where,*$\Delta M\left(T_{i},T_{j}\right)\in \left[0,1\right]$.*By defining a trajectory similarity threshold*$\sigma_{s}$*, a set of synchronized trajectories is said to be a similar trajectory set*$S_{s}$*if the SM difference between any two trajectories in the trajectory set is smaller or equal to*$\sigma_{s}$.

**Definition** **5.***Trajectory Graph* [19]. *A trajectory graph is formed by a set of synchronous trajectories as a weighted undirected graph*
TG=(V,E,W)*, where*
V
*is the set of vertexes in which a vertex*
$v_{i}$
*represents a trajectory*
$T_{i}$. E
*is the set of edges in which an edge*
$e_{i,j}$
*exists between vertexes*
$v_{i}$
*and*
$v_{j}$
*when*
$T_{i}$
*is similar with*
$T_{j}$. W
*is the set of the weight of edge*
E
*where*
$w_{i,j}$
*is the SM difference between*
$T_{i}$
*and*
$T_{j}$.
*A graph is called a clique when there is an edge between each pair of vertices of the graph. A clique with k vertices is called a k-clique [19].*


**Definition** **6.**
*Trajectory Privacy Disclosure Probability. Suppose the anonymity set*
$S_{s}$
*is sent to the location services provider. The attack similarity threshold that the attacker can distinguish the fake trajectory in the anonymity set is*
$\sigma_{a}$
*. When*
$\sigma_{s}<\sigma_{a}$
*, any two trajectories of the set are similar to the attacker. The attacker cannot distinguish any fake trajectory in the set. When*
$\sigma_{s}>\sigma_{a}$
*, suppose the mobility difference*
$\Delta M\left(T_{i},T_{j}\right)$
*between*
$T_{i}$
*and*
$T_{j}$
*in the set is greater than*
$\sigma_{a}$
*, the two trajectories are not similar to the attacker. A trajectory is easier to be distinguished by the attacker when it has fewer similar trajectories.*
*Suppose the trajectory graph*TG=(V,E,W)*is constructed by a set of*k*synchronous trajectories. Let the trajectory graph*$TG_{s}=\left(V,E_{s},W\right)$*be determined by*$\sigma_{s}$*. According to Definition 5, the value of*$\left|E_{s}\right|$*is calculated as*$\frac{k\left(k-1\right)}{2}$*. Let the trajectory graph*$TG_{a}=\left(V,E_{a},W\right)$*be determined by*$\sigma_{a}$*, the degree of vertex*$v_{i}$*in*$TG_{a}$*is*$d_{i}$*. The trajectory is distinguished easily by the attacker when*$d_{i}$*is small. Let the sum of the degrees of all the vertices of*$TG_{a}$*is*$\left|E_{a}\right|$. *When*$\left|E_{a}\right|$*is small, the fake trajectories are more likely to be distinguished by the attacker. Thus, the trajectory privacy disclosure probability is greater. Therefore, the trajectory privacy disclosure probability is defined as follows:*(4)$$P=1-\frac{\left|E_{a}\right|}{\left|E_{s}\right|}.$$

## 4. Spatiotemporal Mobility (SM) based Trajectory Privacy-Preserving Algorithm (MTPPA)

In this section, the overview of the proposed MTPPA algorithm is revealed in Figure 2. There are three stages in MTPPA. In stage I, the trajectory pre-processing is designed. The equivalence classes are formed, the stopovers are detected. In stage II, the process of initial trajectory candidate selection and the construction of trajectory graph is designed. In stage III, an optimal trajectory k-anonymity set is selected by the simulated annealing algorithm. After passing all three stages, the constructed optimal anonymity set can protect the user’s trajectory privacy while matching the requirements of high-quality services.

### 4.1. Trajectory Pre-Processing

The operations in stage I is similar to the processes for handling trajectories in the equivalent classes and in Huo et al.’s method [20]. The pre-processing includes a process for detecting the stopovers in the trajectory. Different from Huo et al.’s method, we consider protecting the trajectory privacy through hiding stopovers in the trajectory in this work.

To guarantee the fake trajectories formed by the remaining dummy stopovers are reachable in the given request time interval [8], the equivalent trajectory time interval $\left[t_{s},\;t_{e}\right]$ is generated according to the initial sampling time $t_{1}$ and the last sampling time $t_{n}$. Moreover, an initial timestamp ${t_{s}}^{\prime }$ according to the timestamp of historical trajectory $T_{h}$ is selected before an end timestamp ${t_{e}}^{\prime }$ is selected such that:(5)$${t_{e}}^{\prime }-{t_{s}}^{\prime }=t_{e}-t_{s}.$$

To keep the computation simple, all the sampling points of the historical trajectory are replaced by the real trajectory timestamps generated according to the user’s speed. Thus, the equivalence class is formed by both the real trajectory and the historical trajectory data.

For the equivalence class formed by the real trajectory $T_{r}$ and the historical trajectory $T_{h}$, if there is a sampling time in $T_{r}$ but not in $T_{h}$, a new sampling point $\left(x_{i},\;y_{i},\;t_{i}\right)$ is inserted at $T_{h}$. Contrary, if there is a sampling time $t_{i}$ in $T_{h}$ but not in $T_{r}$, remove $t_{i}$ from $T_{h}$.

After synchronizing the trajectories, the detection process is used to find the stopovers in the trajectory equivalence class [21]. In practice, we implement DBSCAN algorithm to detect the stopovers. DBSCAN is a popular unsupervised data clustering algorithm. The user needs to predefine the radius r of the stopover. The radius r not only determines the number of stopovers of the trajectory, but also affects the SM of the user. 

### 4.2. Initial Trajectory Candidates Selection

In this stage, the user sets a trajectory similarity threshold $\sigma_{s}$ according to his/her privacy tolerance. It selects 2k−1 trajectories from the trajectory database so that the SM difference between any trajectory and the real trajectory is not greater than $\sigma_{s}$. The selected 2k−1 trajectories and the real trajectories form the initial trajectory candidates set (TC). A weighted undirected trajectory graph model (TG) is used to present the relationship between trajectories and TC. The procedure for constructing the trajectory graph is revealed in Algorithm 1.
**Algorithm 1.** Trajectory Graph Construction (TGC)
**Input:** Initial trajectory candidates set TC, trajectory similarity threshold $\sigma_{s}$.
**Output:** Trajectory graph TG=(V,E,W)
1: $V\leftarrow T_{r},\;E\leftarrow \emptyset ,\;W\leftarrow \emptyset$;
2: $V_{left}\leftarrow TC-V$;
3: **while**
$V_{left}\neq \emptyset$
**do**
4:   **for** each vertex $T_{i}$ in V
**do**
5:     **for** each vertex $T_{j}$ in $V_{left}$
**do**
6:       **if**
$\Delta A\left(T_{i},T_{j}\right)\leq \sigma_{s}\;$
**then**
7:         $w_{i,j}\leftarrow \;s\left(T_{i},T_{j}\right)$;
8:         $E\leftarrow E\cup^{\;}\left(T_{i},\;T_{j},\;w_{i,j}\right)$;
9:         $V\leftarrow V\cup^{\;}T_{j}$;
10:         $W\leftarrow W\cup^{\;}w_{i,j}$;
11:         $V_{left}\leftarrow V_{left}-T_{j}$;
12:       **end if**
13:     **end for**
14:   **end for**
15: **end while**
16: **return**
TG=(V,E,W);

### 4.3. Optimal Anonymization Set Selection

The following process is used to select the optimal k-anonymity set (KAS) from TC. The privacy protection performance of a trajectory anonymity set can be measured by the similarity of anonymity sets. When the k trajectories are similar to each other, the sum of the mobility differences between any two trajectories is as small as possible, the privacy preserving performance is better. Therefore, the problem of finding the optimal k-anonymity set is transformed into the problem of finding the k-clique of an undirected weighted graph [22,23]. It is an NP-hard problem. The process is divided into two parts and explained as follows.

The first part is to search the maximum clique (MC) that contains the vertex of the user’s real trajectory in TG. The number of vertices of MC should be greater than k. To find MC, a greedy algorithm is designed in this paper. It starts with the real trajectory vertex, grow the current clique one vertex at a time by looping through the remaining vertices of the graph. For each vertex v examined by this loop, if v is adjacent to every vertex that is already in the clique, add v to the clique. Otherwise, discard v. The process is shown in Algorithm 2.
**Algorithm 2.** Search for Maximum Clique (SMC)
**Input:** Initial Trajectory graph TG=(V,E,W)
**Output:** Maximum clique $MC=\left(V_{MC},E_{MC},W_{MC}\right)$
1: $V_{MC}\leftarrow T_{r}$;
2: $V_{left}\leftarrow V-V_{MC}$;
3: **for** each vertex $T_{i}$ in $V_{left}$
**do**
4:   **for** each vertex $T_{j}$ in $V_{MC}$
**do**
5:     **if**
$T_{i}$ is adjacent to $T_{j}$
**then**
6:       $V_{MC}\leftarrow V_{MC}\cup^{\;}T_{i}$;
7:       $E_{MC}\leftarrow E_{MC}\cup^{\;}\left(T_{i},\;T_{j},\;w_{i,j}\right)$;
8:       $W_{MC}\leftarrow W_{MC}\cup^{\;}w_{i,j}$;
9:       $V_{left}\leftarrow V_{left}-T_{i}$;
10:     **end if**
11:   **end for**
12: **end for**
13: **return**
MC

The second part is to select k vertices with a smaller sum of weights from MC. This process can be treated as a combinatorial optimization problem. The objective function f(X) of the optimization problem is the sum of the SM differences between k trajectory pairs. X is the decision variable of f(X). $X=\left\{T_{1},T_{2},\ldots ,T_{k}\right\}$. The mathematical model of the objective function is defined as follows:(6)$$\begin{array}{c} \{\begin{array}{c} \min f\left(X\right)=\overset{k}{\underset{i=1}{\sum }}\overset{k}{\underset{j=i+1}{\sum }}\Delta M(T_{i},T_{j})\\ s.t.\;k\leq \left|MC\right|\\ \;T_{i}\in MC\\ \;T_{j}\in MC \end{array}. \end{array}$$

When k is very large, it is hard to find KAS for conventional algorithms in polynomial time. Nevertheless, the heuristic swarm intelligence algorithms are capable of solving the problem with satisfactory efficiency [24]. One of the classical swarm intelligence algorithms for solving this combinatorial optimization problem is the simulated annealing algorithm [25]. It searches the approximate optimal solution more quickly and has strong global searchability. Hence, the simulated annealing algorithm is used to solve the optimization problem (see Algorithm 3).
**Algorithm 3.** Search for k-anonymity set (SKAS)**Input:** Maximum clique MC, k, initial temperature $TP_{0}$, minimum temperature $TP_{min}$, times of internal circulation of every temperature L.
**Output:**
k-anonymity set KAS
1: Select k Trajectories randomly from MC, set to $X_{0}$;
2: $X^{*}\leftarrow X_{0}$;
3: Calculate $f\left(X^{*}\right)$;
4: t←0;
5: **while**
$TP\geq TP_{min}$
**do**
6:   **for**
i=0;i<L;i++
**do**
7:     Select k Trajectories randomly from MC, set to $X_{new}$;
8:     Calculate $f\left(X_{new}\right)$;
9:     **if**
$f\left(X_{new}\right)<\;f\left(X^{*}\right)$
**then**
10:       $X^{*}\leftarrow X_{new}$;
11:     **else**
12:       $p\leftarrow \exp\left(-\frac{f\left(X_{new}\right)-f\left(X^{*}\right)}{TP}\right)$;
13:       **if**
random(0,1)< p
**then**
14:         $X^{*}\leftarrow X_{new}$;
15:       **end if**
16:     **end if**
17:   **end for**
18:   t++;19:   TP←TP01+t;20: **end while**
21: KAS←X*;22: **return**
KAS

Algorithm 3 can be summarized in four steps:

(1)Set the initial high-temperature $TP_{0}$, minimum temperature $TP_{min}$, the number of iterations L for each temperature TP.(2)Select an initial solution $X_{0}$ randomly from MC. Let $X_{0}$ be the optimal solution $X^{*}$. Calculate $f\left(X^{*}\right)$.(3)Repeat L iterations for each temperature TP. For each temperature, generate a new solution $X_{new}$, if $f\left(X_{new}\right)<f\left(X^{*}\right)$. Then, let $X^{*}=X_{new}$. Otherwise, the optimal solution will accept $X_{new}$ at a probability p. It follows the Metropolis criterion and decreases with the decrease of temperature TP. The criterion is shown as follows.
(7)$$p=\{\begin{array}{c} 1,\;if\;f\left(X_{new}\right)<f\left(X^{*}\right)\\ \exp\left(-\frac{f\left(X_{new}\right)-f\left(X^{*}\right)}{TP}\right),\;if\;f\left(X_{new}\right)\geq f\left(X^{*}\right) \end{array}$$(4)Gradually reduce the temperature TP. End the process until TP is less than $TP_{min}$. Then return $X^{*}$. The temperature reduction mode is as follows.
(8)$$TP\left(t\right)=\frac{TP_{0}}{1+t}$$

## 5. Experiment

The implementation details, the feasibility analysis, the data availability analysis and security analysis results are reported in this section. The implementation details are described in Section 5.1. A case study is provided to demonstrate the MTPPA in the feasibility analysis in Section 5.2. The data availability analysis is discussed in Section 5.3. The security analysis is discussed in Section 5.4.

### 5.1. Implementation Details

The experiment was implemented with PyCharm in Python 3.8 on a Windows 10 operating system with Intel(R) Core(TM) i3-7100U @ 2.40 GHz equipped with 4 GB RAM. The algorithms were repeated 50 times to ensure the results obtained with different variable values were stable. The experiment uses the user’s trajectory obtained from Microsoft’s GeoLife Trajectories 1.3 [6] as the historical trajectory. This dataset contains 17,621 trajectories, recording a wide range of outdoor activities of users’ daily life, such as going home, going to work, shopping, and dining. The travel modes include driving, by bus, by train, by bicycle, and walking. This dataset has been applied to mobile pattern mining, location-based social network, location privacy, and location recommendation. After trajectory pre-processing, each trajectory of the dataset contains a sequence of 20 sampling points. That is n=20. 3200 trajectories are selected randomly to form a trajectory equivalence class as the experimental data. After the trajectory pre-processing, the proportion of the trajectories with the number of stopovers is 1 is the largest when r = 0.1. 

To verify the effectiveness of the proposed algorithm, α and β were set as 0.9 and 0.1, respectively. V=10km/h, r = 0.1 km, $\left|V_{MC}\right|=1.3k$. The data availability was measured by the information loss. Less information loss means better data availability. The information was estimated by the size of cloaking area, similar to Hu et al.’s method [26]. The trajectory privacy disclosure probability of the algorithm was analyzed. The proposed MTPPA algorithm was compared with the DTI algorithm [26] and the random algorithm [8]. The DTI-1 algorithm represents the case when the DTI algorithm only considers data utility. The DTI-2 algorithm refers to the case when the DTI algorithm only considers trajectory privacy. The Random algorithm selects the k-anonymity set randomly in the trajectory candidates set. 

### 5.2. Feasibility Analysis

To explain the application of MTPPA more clearly, a simple case study is given to describe the selection process of trajectory anonymity set when k=4, $\sigma_{s}=0.3$, $\left|V_{MC}\right|=1.3k$, r=0.1.

As shown in Figure 3a, the initial trajectory candidates map is constructed by a real trajectory $T_{r}$ and seven historical trajectories $T_{1}$, $T_{2}$, $T_{3}$, $T_{4}$, $T_{5}$, $T_{6}$, $T_{7}$. The number of stopovers and the average moving speed of these trajectories are listed in Table 2, where the SM of each trajectory is computed by Equation (2). The parameters are set as α=0.9, β=0.1, V=10km/h. Equation (3) is used to calculate the SM difference between trajectories. The weight matrix of the eight trajectories is obtained as follows:$$\begin{array}{cccccccc} \;\;\;\;\;\;\;\;\;\;\;\;\;\;\;T_{r}\;\;\;\;\; & T_{1}\;\;\;\;\; & T_{2}\;\;\;\;\; & T_{3}\;\;\;\;\; & T_{4}\;\;\;\;\; & T_{5}\;\;\;\;\; & T_{6}\;\;\;\;\; & T_{7} \end{array}\;W=\begin{array}{c} T_{r}\\ T_{1}\\ T_{2}\\ T_{3}\\ T_{4}\\ T_{5}\\ T_{6}\\ T_{7} \end{array}\left[\begin{array}{cccccccc} 0 & 0.222 & 0.116 & 0.152 & 0.188 & 0.062 & 0.081 & 0.208\\ 0.222 & 0 & 0 & 0.070 & 0.034 & 0.284 & 0.141 & 0\\ 0.116 & 0 & 0 & 0.268 & 0 & 0.055 & 0.197 & 0.092\\ 0.152 & 0.070 & 0.268 & 0 & 0.036 & 0.214 & 0.071 & 0\\ 0.188 & 0.034 & 0 & 0.036 & 0 & 0.250 & 0.108 & 0\\ 0.062 & 0.284 & 0.055 & 0.214 & 0.250 & 0 & 0.142 & 0.147\\ 0.081 & 0.141 & 0.197 & 0.071 & 0.108 & 0.142 & 0 & 0.289\\ 0.208 & 0 & 0.092 & 0 & 0 & 0.147 & 0.289 & 0 \end{array}\right]$$
where the weight of two trajectories is 0, which means that the two trajectories are not similar.

Figure 3b is the initial trajectory candidates graph constructed by the weight matrix of the eight trajectories. The maximum clique MC is obtained by Algorithm 3. It contains six trajectories $T_{r}$, $T_{1}$, $T_{3}$, $T_{4}$, $T_{5}$, $T_{6}$. By using algorithm 4, four trajectories with the smallest sum of weights $T_{r}$, $T_{3}$, $T_{4}$, $T_{6}$ are found from MC to form the optimal anonymity set.

### 5.3. Data Availability Analysis

In this subsection, the comparison of algorithms in terms of information loss with different value of *k* is revealed. 

Figure 4a shows the information loss comparison between these four algorithms with k increases when $\sigma_{s}=0.2$, $\sigma_{a}=0.1$, N=8. As shown in Figure 4a, the information loss of the four algorithms decreases when k increases. For the same k, the information loss of the random algorithm and the DTI-2 algorithm are relatively high. The information loss of the DTI-1 algorithm and the MTPPA algorithm are similar, but both of them are relatively lower than the random algorithm and the DTI-2 algorithm. This is because the DTI-1 algorithm essentially does not consider the similarity between trajectories. The k-anonymity set generated by the MTPPA algorithm contains the user’s real trajectory and k − 1 historical trajectories. The queried results contain the query results of the user’s real location in each query. Therefore, the MTPPA algorithm and the DTI-1 algorithm have the lower information loss, which results in better data availability.

### 5.4. Security Analysis

In this subsection, the comparison of algorithms in terms of trajectory privacy disclosure probability with different value of *k*, N, $\sigma_{a}$, $\sigma_{s}$ are revealed, respectively. 

#### 5.4.1. Comparison of Algorithms in Terms of Trajectory Privacy Disclosure Probability under Different *k*

Figure 4b shows the trajectory privacy disclosure probability comparison between these four algorithms with k increases when $\sigma_{s}=0.2$, $\sigma_{a}=0.1$, N=8. As shown in Figure 4b, the trajectory privacy disclosure probability of the four algorithms remains unchanged. For the same k, both of the trajectory privacy disclosure probability of the Random algorithm and the DTI-1 algorithm are relatively high. The trajectory privacy disclosure probability of the DTI-2 algorithm is lower, and that of the MTPPA algorithm is lower than that of the DTI-2 algorithm by 37%. This is because the random algorithm and the DTI-1 algorithm essentially do not consider the similarity between trajectories. Therefore, the random algorithm and the DTI-1 algorithm have a high probability of privacy disclosure. The DTI-2 algorithm considers the similarity between trajectories, but it does not guarantee the final k-anonymity set is a similar trajectory set. The MTPPA algorithm guarantees the final k-anonymity set is a similar trajectory set. Therefore, the MTPPA algorithm has the lowest trajectory privacy disclosure probability.

#### 5.4.2. Comparison of Algorithms in Terms of Trajectory Privacy Disclosure Probability under Different N

Figure 4c shows the trajectory privacy disclosure probability comparison between these four algorithms in the condition that N increases when $\sigma_{s}=0.2$, $\sigma_{a}=0.1$, k=6. It can be observed from Figure 4c that with N increases, the trajectory privacy disclosure probability of the DTI algorithm and the Random algorithm slightly increases. For any N value, the proposed MTPPA algorithm still has the lowest trajectory privacy disclosure probability, which is 42% lower than that of the DTI-2 algorithm. All of the four algorithms have the lowest trajectory privacy disclosure probability when N=1. This is because in the selected experimental trajectories, the proportion of the trajectories with the number of stopovers is the largest. When selecting the initial trajectory candidates, the probability of selecting these trajectories is higher. Thus, the trajectories of the final k-anonymity set are more similar.

#### 5.4.3. Comparison of Algorithms in Terms of Trajectory Privacy Disclosure Probability under Different $\sigma_{a}$

Figure 4d shows the trajectory privacy disclosure probability comparison between these four algorithms with $\sigma_{a}$ increases when $\sigma_{s}=0.2$, N=8, k=6. It can be observed from Figure 4d that the trajectory privacy disclosure probability of the four algorithms decrease when $\sigma_{a}$ increases. The trajectory privacy disclosure probability of the Random algorithm is the highest, while that of the MTPPA algorithm is the lowest. When $\sigma_{a}=0,$ the trajectory privacy disclosure probability of the four algorithms is 1. In the view of the attacker, all the trajectories of the k-anonymity set are dissimilar. When $\sigma_{a}=0.2$, the trajectory privacy disclosure probability of the MTPPA algorithm is 0. This is because at this time, $\sigma_{s}=\sigma_{a}$. From the attacker’s point of view, the trajectories of the k-anonymity set generated by the MTPPA algorithm is similar to each other. The trajectory privacy of the k-anonymity set generated by the other algorithms is still at the risk of disclosure.

#### 5.4.4. Comparison of Algorithms in Terms of Trajectory Privacy Disclosure Probability under Different $\sigma_{s}$

Figure 4e shows the trajectory privacy disclosure probability comparison between these four algorithms with $\sigma_{s}$ increases when $\sigma_{a}=0.1$, N=6, k=6. It can be observed from Figure 4e that the MTPPA algorithm has the lowest trajectory privacy disclosure probability. When $\sigma_{s}$ is smaller than $\sigma_{a}$, the trajectory privacy disclosure probability of the MTPPA algorithm is 0. When $\sigma_{s}$ is greater than $\sigma_{a}$, with $\sigma_{s}$ increases, the trajectory privacy disclosure probability of the MTPPA algorithm gradually increases. Although the trajectory similarity threshold $\sigma_{s}$ of the maximum clique generated by the MTPPA algorithm is very close to $\sigma_{a}$. Most of the SM difference between trajectories is far less than $\sigma_{a}$. When $\sigma_{s}$ continue to increase, less and less of the SM difference between trajectories is smaller than $\sigma_{a}$. As a result, the trajectory privacy disclosure probability is increasing.

## 6. Conclusions

In this paper, spatiotemporal mobility (SM) is defined to measure the similarity between trajectories. The relationship between the SM and the anonymity set is discovered. The mathematical model is constructed to model the relationship between trajectories. Based on SM and trajectory graph modeling, the MTPPA algorithm is proposed. The problem of finding the optimal k-anonymity set is transformed into the k-clique problem of an undirected weighted graph. The simulated annealing algorithm is utilized to find the approximate optimal k-anonymity set. It improves the similarity between trajectories of the anonymity set effectively while meeting the same services quality. Experimental results show that the trajectory privacy disclosure probability of the k-anonymity set generated by this algorithm is about 40% lower than that of existing algorithms.

This study considers the privacy protection effect when the historical trajectories are sufficient, but not the case when the historical trajectories are sparse. Future studies may concentrate on the following aspects: (1) The privacy protection effect of this algorithm will be discussed under the condition of the historical trajectories are sparse. (2) Based on the SM, the semantic information of the stopover will be considered to achieve semantically secure anonymity. (3) The model and algorithm designed in this paper cans be applied to popular services such as online car-hailing to match the best vehicle for the users without disclosing sensitive information of the users and the drivers.

## Figures and Tables

**Figure 1 sensors-21-02021-f001:**
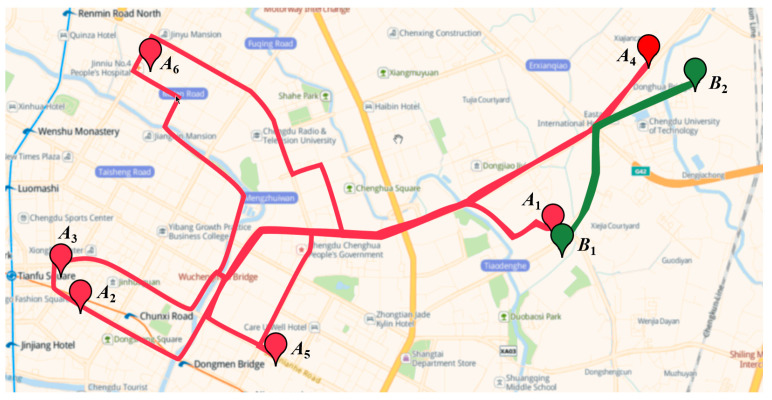
Moving trajectories of different users.

**Figure 2 sensors-21-02021-f002:**
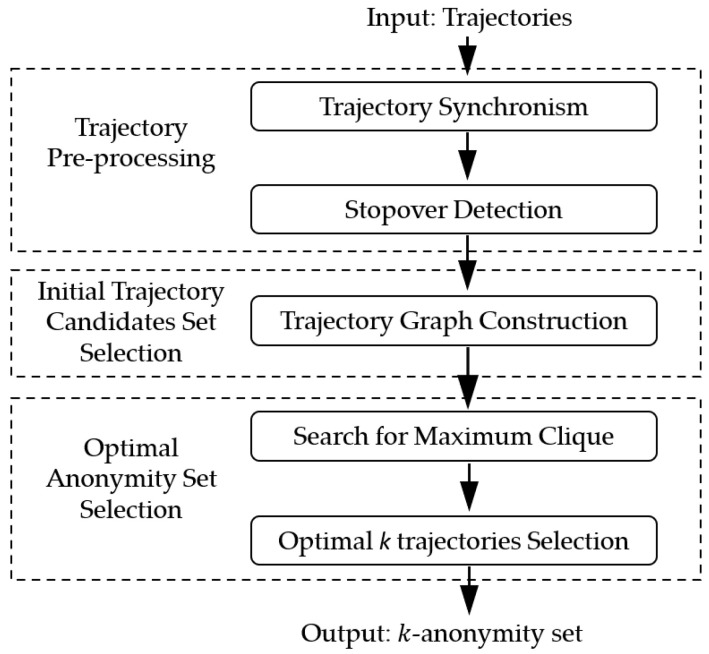
Overview of the proposed algorithm.

**Figure 3 sensors-21-02021-f003:**
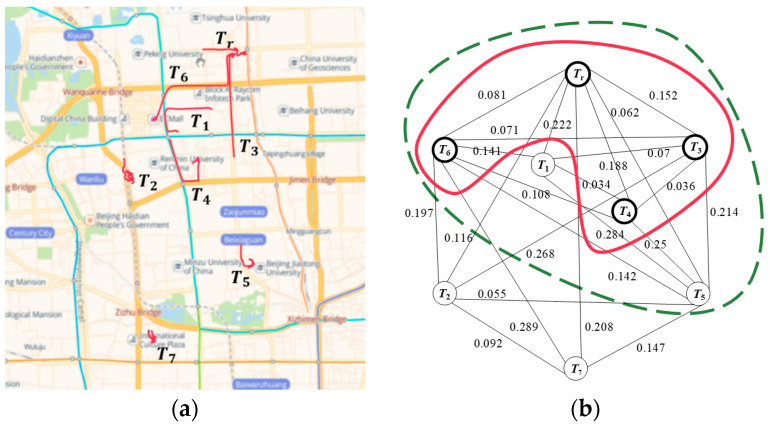
A case study. (**a**) Initial trajectory candidate map. (**b**) Initial trajectory candidate graph.

**Figure 4 sensors-21-02021-f004:**
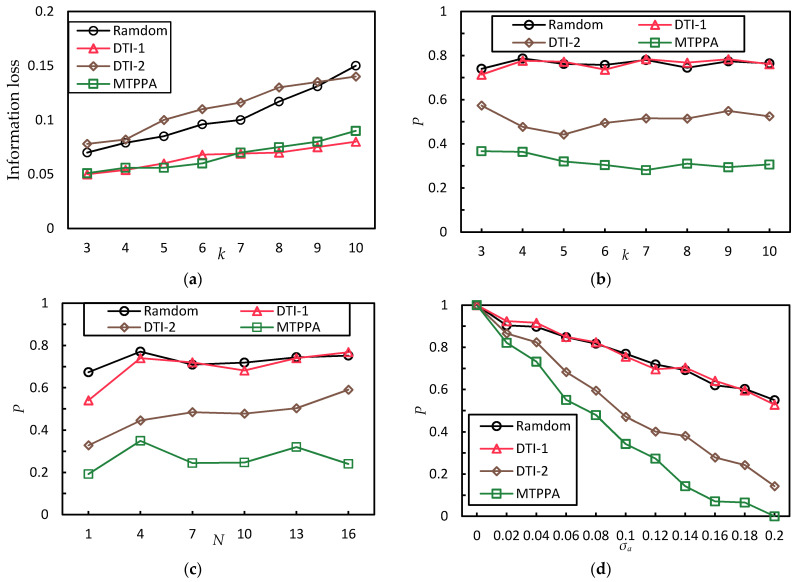
Comparison of algorithms under different parameters. (**a**) Comparison of algorithms in terms of information loss under different *k*. (**b**) Comparison of algorithms in terms of trajectory privacy disclosure probability under different *k*. (**c**) Comparison of algorithms in terms of trajectory privacy disclosure probability under different N. (**d**) Comparison of algorithms in terms of trajectory privacy disclosure probability under different $\sigma_{a}$. (**e**) Comparison of algorithms in terms of trajectory privacy disclosure probability under different $\sigma_{s}$.

**Table 1 sensors-21-02021-t001:** Notation.

Symbols	Definitions
k	Anonymity degree
$t_{s}$	Starting timestamp of a trajectory
$t_{e}$	Ending timestamp of a trajectory
M	Spatiotemporal Mobility
α	The proportion of the number of stopovers of the spatiotemporal mobility
β	The proportion of the average moving speed of the spatiotemporal mobility
N	The number of stopovers
n	The number of sampling points of the trajectory
$\bar{v}$	The average moving speed
$v_{max}$	The maximum speed limit of the anonymous region
$\Delta M\left(T_{i},T_{j}\right)$	The mobility difference between $T_{i}$ and $T_{j}$
V	The set of vertexes of trajectory graph
E	The set of edges of trajectory graph
W	The set of the weight of edge E
$\sigma_{s}$	Trajectory similarity threshold
d	The degree of vertex
$\sigma_{a}$	Attack similarity threshold
P	Trajectory Privacy Disclosure Probability

**Table 2 sensors-21-02021-t002:** Number of stopovers and average moving speed of the initial trajectory candidates.

Trajectory	Number of Stopovers	Average Moving Speed	Spatiotemporal Mobility
$T_{r}$	6	3.966	0.3097
$T_{1}$	11	3.663	0.5316
$T_{2}$	4	1.36	0.1936
$T_{3}$	9	5.667	0.4617
$T_{4}$	10	4.805	0.2481
$T_{5}$	5	2.309	0.3903
$T_{6}$	7	7.527	0.1014
$T_{7}$	2	1.135	0.498

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
