# Peer review of "Spatiotemporal Mobility Based Trajectory Privacy-Preserving Algorithm in Location-Based Services"

_sensors, 2021, doi:10.3390/s21062021_

Round 1
Reviewer 1 Report
This is an extremely difficult to read paper. It is very long, at 17 pages and most of the text requires major rewrite because of the bad use of english. Examples:
"It is assumed that time and space are discretedata."
"should not less than ?"
Take the abstract, "user activeness" appears in it 5 times. I disagree with the name itself as it just sounds bad, but that is the authors choice.
It is not appropriate to add market estimations in a scientific article. This is also irrelevant, as location-based services importance is uncontested. It is also not acceptable to cite a website for this kind of information.
E is not defined.
The experiments show only how effective the privacy preserving proposal is. It doesn't say anything about how useful the data is after it has been modified to respect privacy.
Reviewer 2 Report
Dear Authors,
The manuscript is excellent, and well-written overall. The manuscript has sufficient originality, and undertaken problem is very recent. Although the results presented in the manuscript seem promising and overall approach is contributing significantly in the body of the literature, I encourage the authors to please consider the attached file suggestions to improvise the presented work more. Thanks

Reviewer 3 Report
This manuscript investigates the problem of using k-anonymity based methods to protect the users’ trajectory privacy. It mainly considers to improve the privacy protection effect of the method from the new perspective of user activeness, and use trajectory graph to model the relationship between trajectories. This manuscript is innovated, the motivation behind the problem investigated in this manuscript is interesting and meaningful. Just a few concerns:
- In the feasibility analysis, in order to show the differences of the trajectory graph in different trajectory-selecting processes, can the maximum clique and the optimal k-anonymity set be clearly identified in the trajectories?
- In the experimental analysis, the format of the parameters in the subtitle should be consistent. I suggest that the font of these parameters should be uniformly not bold.
- As a minor comment, I suggest the authors to do a careful proofreading to correct the typos along the manuscript.
Round 2
Reviewer 1 Report
Most importantly: This paper is far too long, has too many pages, too many references, too many figures. This makes the paper very difficult to read.
Although many corrections have been made, the over-all English style and language needs improvement.
I would suggest to the authors to cut as much from the paper without loosing the essence and then concentrate on improving the english.
To clarify what I mean about improving the english. And I insist that these are just examples, there are many similar cases throughout the paper:
The first sentence from the abstract: "Location-Based Services (LBSs) facilitates our daily life greatly"
facilitate - although gramatically correct, better fits an action "facilitates locomotion"
greatly - why add this? no need to exagerate the importance, and I would argue that although important, and LBS-s do not greatly improve out life.
The next sentence: "Intelligent services are provided by LBSs through collecting locations and trajectories data of users, which potentially exposes the privacy issues leaking of users’ trajectories easily"
Intelligent services are not provided through data collection. There is nothing intelligent about data collection. Using data to extract information can lead to services which provide intelligent decisions.
LBSs only collect location data, from location data we can build trajectories. One does not collect trajectory data.
The sentence is too long. You can put a period after users.
The next part is very confusing. You don't mean to say this exposes privacy issues. It's the other way around, exposure of data can be a privacy issue. Maybe exposing is not the correct word. After re-reading many times I though the authors meant is "potentially raises privacy issues".
There is nothing connecting leaking of trajectories to the previous part of the sentence.
There is no need to add the word "easily". Even if it is required it shouldn't be the last word in the sentence, this is not where it goes in English.
